# BBIBP-CorV Vaccination against the SARS-CoV-2 Virus Affects the Gut Microbiome

**DOI:** 10.3390/vaccines11050942

**Published:** 2023-05-04

**Authors:** Yang Shen, Ying Dong, Jie Jiao, Pan Wang, Mulei Chen, Jing Li

**Affiliations:** 1Department of Nephrology, Beijing Chaoyang Hospital, Capital Medical University, Beijing 100020, China; 2Heart Center and Beijing Key Laboratory of Hypertension, Beijing Chaoyang Hospital, Capital Medical University, Beijing 100020, China; 3Department of Cardiology, Beijing Chaoyang Hospital, Capital Medical University, Beijing 100020, China

**Keywords:** SARS-CoV-2, coronavirus disease 2019 (COVID-19), vaccines, gut, microbiome

## Abstract

Several observational studies have confirmed that the severe acute respiratory syndrome coronavirus2 (SARS-CoV-2) might substantially affect the gastrointestinal (GI) system by replicating in human small intestine enterocytes. Yet, so far, no study has reported the effects of inactivated SARS-CoV-2 virus vaccines on gut microbiota alterations. In this study, we examined the effects of the BBIBP-CorV vaccine (ChiCTR2000032459, sponsored by the Beijing Institute of Biological Products/Sinopharm), on gut microbiota. Fecal samples were collected from individuals whoreceived two doses of intramuscular injection of BBIBP-CorV and matched unvaccinated controls. DNA extracted from fecal samples was subjected to 16S ribosomal RNA sequencing analysis. The composition and biological functions of the microbiota between vaccinated and unvaccinated individuals were compared. Compared with unvaccinated controls, vaccinated subjects exhibited significantly reduced bacterial diversity, elevated firmicutes/bacteroidetes (F/B) ratios, a tendency towards Faecalibacterium-predominant enterotypes, and altered gut microbial compositions and functional potentials. Specifically, the intestinal microbiota in vaccine recipients was enriched with Faecalibacterium and Mollicutes and with a lower abundance of Prevotella, Enterococcus, Leuconostocaceae, and Weissella. Microbial function prediction by phylogenetic investigation of communities using reconstruction of unobserved states (PICRUSt) analysis further indicated that Kyoto Encyclopedia of Genes and Genomes (KEGG) pathways involved in carbohydrate metabolism and transcription were positively associated with vaccine inoculation, whereas capacities in neurodegenerative diseases, cardiovascular diseases, and cancers were negatively affected by vaccines. Vaccine inoculation was particularly associated with gut microbiota alterations, as was demonstrated by the improved composition and functional capacities of gut microbiota.

## 1. Introduction

COVID-19 is considered one of the worst pandemics in human history. So far (April 2023), it has killed more than 6.8 million people and infected approximately 686 million people worldwide [1,2]. Lungs are the primary target organ of the virus, causing severe acute fever, dry cough, fatigue, and dyspnea; however, the virus may also affect extrapulmonary sites in other organs, including the GI tract [3]. Several observational studies successively confirmed a substantial involvement of GI systems, including the ability of the virus to infect and replicate in human small intestine enterocytes [4]. In addition, positive detection of viral RNA has been found in fecal samples using RT-polymerase chain reaction (PCR) and electron microscopy [5,6,7]. The gut microbiome is an essential part of the GI tract; it has a prominent role in body metabolism, immune response modulations, functional homeostasis [8,9], and works as a defensive system against malignant infections [10]. Researchers have made efforts to decode the bidirectional relationship between the gut microbiome and COVID-19. It has been suggested that disturbances in gut composition increase the susceptibility to COVID-19 [11]. Compared to non-COVID-19 individuals, the gut microbiota composition of COVID-19 patients has been reported to be different, with extremely lower gut microbial richness [12] and dramatic depletion of bacteria with immunomodulatory potentials [13]. Facing the emergency situation of rapid transmission of the SARS-CoV-2 infection, an urgent demand for developing vaccines targeting the virus to curb and protect against COVID-19-associated mortality has emerged. According to the World Health Organization’s draft landscape of COVID-19 candidate vaccines, 183 candidate vaccines, including 22 inactivated SARS-CoV-2 virus vaccines, are currently under clinical evaluations and 199 candidate vaccines are in the stage of preclinical assessments [14]. Out of the inactivated SARS-CoV-2 virus vaccines, BBIBP-CorV (ChiCTR2000032459), sponsored by the Beijing Institute of Biological Products/Sinopharm, has been approved for emergency use. A phase I/II clinical trial investigating BBIBP-CorV [15] suggested that patients may develop a humoral response against SARS-CoV-2 four days after the first inoculation. Moreover, a 100% seroconversion has been observed in all the participants forty-two days after vaccine administration. The overall adverse reactions were mild or moderately severe and well tolerated by most healthy individuals. In addition, the safety and efficacy of the Sinopharm vaccine (BBIBP-CorV) have been verified in both the elderly population and children [16,17].

Considering significant impairments and dysbiosis occurring in the intestinal flora of COVID-19 patients, it is plausible to think that vaccine administration against the SARS-CoV-2 virus may alleviate GI symptoms, improve gut dysbiosis, and enhance gut homeostasis possibly by selectively stimulating certain members of the gut ecosystem. A correlation of gut microbiota and metabolic functions with the antibody response to the BBIBP-CorV vaccine has been recently documented. It was found that several short-chain fatty acids displayed a positive correlation with the antibody response [18]. Meanwhile, preexisting antibodies targeting SARS-CoV-2 S2 were demonstrated to cross-react with gut bacteria and further impact the immunity induced by the COVID-19 vaccine [19]. It was inferred that gut microbiota possibly play a role in influencing the immune responses to COVID-19 vaccinations via mechanisms that include the effects of lipopolysaccharides, flagellin, peptidoglycan, and short-chain fatty acids [20]. In the current study, investigators randomly enrolled 20 healthy adults who received two doses of intramuscular injection of BBIBP-CorV (vaccinated group) and 20 healthy adults who were not vaccinated (unvaccinated group) to characterize and describe the features and profiles of the intestinal bacteria following SARS-CoV-2 virus vaccine injections and further explore the reciprocal effect between the vaccines and gut microbiota. These findings offer important implications for future therapeutic vaccine development beyond targeting COVID-19.

## 2. Materials and Methods

### 2.1. Study Design and Participants

In the present study, 40 adults aged 18–59 years and eligible for catch-up vaccination were invited for participation at Beijing Chaoyang Hospital between 1 January 2021 and 1 April 2021. The inclusion criteria were adults who received either the complete two doses of intramuscular injection of BBIBP-CorV (vaccinated group, *n* = 20) or those who were not vaccinated (unvaccinated group, *n* = 20). The vaccination institutions provided a certificate recording the type, dose, injection date, and manufacturer of the vaccines for each individual. The vaccination history of the participants was confirmed according to the vaccination certificate. The exclusion criteria were participants with cancer, previous heart failure, renal failure, stroke, peripheral artery disease, and chronic inflammation disorders; with previous SARS-CoV-2 exposure or infection; and those who received statin, aspirin, insulin, metformin, antibiotics, or probiotic treatments within the last two months. Individuals who had previously caught SARS-CoV-2 infection were strictly excluded in the current study. We confirmed that all the participants were free from SARS-CoV-2 infection by examining their previous nucleic acid testing results for COVID-19 from throat swabs. The study was performed in accordance with the Helsinki declaration and was approved by the Medical Ethics Committee from Beijing Chaoyang Hospital (approval number 2021-ke-440). Written informed consent was obtained from all study participants prior to enrollment.

### 2.2. Fecal Sample Collection and DNA Preparation

The fresh middle section of the fecal samples was collected from all participants within one month following the administration of a second dose of vaccines. All the samples were transported to the laboratory on ice within two hours after collection, frozen, and stored at −80 °C until DNA extraction. The total genomic DNA isolation from the stool samples was performed using the TIANamp Stool DNA Kit (TIANGEN Biotech (Beijing) Co., Ltd., Beijing, China) according to the manufacturer’s instructions. All the prepared DNA samples were quantified and stored at −20 °C prior to further use.

### 2.3. PCR Amplicon and Sequencing of 16S rRNA Gene

The V3-V4 hypervariable region of the 16S rRNA gene was amplified by PCR using the following primers: 338F 5′-ACTCCTACGGGAGGCAGCAG-3′ and 806R 5′-GGACTACHVGGGTWTCTAAT-3′. Subsequent to 16S rRNA library preparation and generation, the library quality was assessed on the Qubit@2.0 Fluorometer (Thermo Scientific, Waltham, MA, USA) and Agilent Bioanalyzer 2100 system. Then, the libraries’ sequencing was conducted on an Illumina HiSeq platform 2500 (Illumina, SanDiego, CA, USA). The raw data were filtered to exclude low-quality reads. CLC Genomics Workbench 9.5.2 (QIAGEN, Denmark, Germany) was used to merge paired-end reads and generate clean tags, which were then filtered according to Quantitative Insights Into Microbial Ecology (QIIME, version 1.7.0). Operational taxonomic units (OTUs) with ≥97% similarity were determined using the Uparse pipeline (Version 7.0.1001) by clustering all the sequences [21]. The representative sequence of each OTU was selected, and the taxonomic information was annotated using the RDP classifier [22] and the GreenGene database at the genus level [23]. The functional capabilities of gut microbial communities were predicted using the PICRUSt Version 1.0.0 according to the KEGG pathway database.

### 2.4. Bioinformatics and Statistical Analysis

The QIIME software (Version 1.7.0) and R package were used for the determination of the microbial α-diversity (variation within the sample) and β-diversity (variation between samples). For α-diversity, indexes including Chao1, Shannon, Pielou, ACE, Simpson, and Invsimpson were calculated at the genus level. For β-diversity, principal component analysis (PCA), principal co-ordination analysis (PCoA), and non-metric dimensional scaling (NMDS) plots were conducted to describe and visualize the microbial similarities and differences between the samples according to the Bray–Curtis distance based on the genera. Analysis of similarities (Anosim) was performed to confirm the microbial difference between groups. To identify the biomarker bacteria between groups, the taxa underwent statistically significant variation and was analyzed by linear discriminant analysis effect size (LEfSe) with a cutoff of |LDA score (log10)| > 2 and *P* value < 0.05. The gut enterotypes were analyzed using the partitioning around medoid method based on the relative abundance of genera. Each enterotype was clustered using the PCA of the Jensen–Shannon distance across all samples and named by the most abundant genera. Quantitative data were presented as median (first quartile, third quartile), and categorical variables were presented as frequencies and percentages. The Wilcoxon rank-sum test, chi-square test, and Kruskal–Wallis test were used to compare the characteristics of study participants as appropriate. A *P* value < 0.05 was considered statistically significant.

## 3. Results

### 3.1. Clinical Characteristics of the Recruited Subjects

We collected fecal samples from 20 adults who received BBIBP-CorV and 20 unvaccinated control subjects. Their information on demographics, physiology, and biochemistry tests is summarized in Table 1. Clinical characteristics between groups were similar, with no differences in age, gender, body mass index (BMI), blood pressure levels, fasting blood glucose, total cholesterol, triglyceride, high-densitylipoprotein, low-density lipoprotein, uric acid, and white blood cells (WBC) (all *P* > 0.05).

### 3.2. Differences in Gut Microbiome Community Structure in Unvaccinated and Vaccinated Subjects

To investigate the global structure of the gut microbiota among unvaccinated and vaccinated subjects, we applied 16S rRNA sequencing to the bacterial DNA isolated from fecal samples on an Illumina HiSeq sequencer, generating 4,373,985 raw data. A total of 4,008,081clean tags were filtered from 4,281,968 raw tags, and 94.67% of all qualified tags were clustered into qualified OTUs by randomly selecting the qualified reads in Appendix A. Finally, 328 qualified OTUs covering almost all sequences were obtained for downstream analysis. We recognized 312 OTUs in the unvaccinated group and 308 OTUs in the vaccinated group; 292 OTUs were shared among the groups.

By examining the number of OTUs within random samples, rarefaction curves were identified as mostly flat, which indicated that the sample size in each group was sufficient and reasonable (Appendix A). A small number of novel OTU characteristics was additionally produced by more sampling. In addition, to further identify whether the sequencing quantity was adequate, another rarefaction curve was performed by gradually expanding the sequencing depth. The flattening of the rarefaction curves demonstrated that the sequencing depth was adequate for covering all the bacterial OTUs in the community (Appendix A).

A comparison of the observed OTUs revealed that recipients receiving vaccines exhibited a significantly reduced bacterial load compared to controls (Figure 1A). Regarding microbiota community diversity, α-diversity analysis was applied to analyze the complexity of genera diversity in each group using several indices. Compared with unvaccinated individuals, the α-diversity indices, as demonstrated by the Chao 1 richness index (*P* = 0.003), the Shannon diversity index (*P* = 0.017), the Pielou evenness index (*P* = 0.062), ACE (*P* = 0.002), and the Simpson (*P* = 0.038) and Invsimpson indices (*P* = 0.039) were significantly lower in the vaccinated group, thus suggesting that the SARS-CoV-2vaccines affected global gut microbiome (Figure 1B–G).

Subsequently, to evaluate the microbiome community structure differences between vaccinated participants and the unvaccinated controls, bacterial β-diversity was evaluated based on the Bray–Curtis distance at the genus level. Significant separations anddistinct clusters of vaccinated subjects and unvaccinated individuals were illustrated using the PCA, PCoA, and NMDS plots (Figure 1H–J; *P* = 0.015, Anosim). The distribution of vaccinated and unvaccinated participants at the axes based on β-diversity plots was assessed using pairwise Wilcoxon tests (Appendix A). Significant disparities were noted at the second PCA (*P* < 0.001) and first NMDS depending on the vaccinated or unvaccinated status. The dissimilarity between the vaccinated subjects and the unvaccinated group strongly reflected the heterogeneity of the gut microbiota signatures upon SARS-CoV-2vaccine administration.

### 3.3. Gut Features and Comparison of Microbial Profiles in Taxa

Taxonomic annotation and abundance profiles of gut bacteria were evaluated in both groups (Figure 2A,C and Appendix A). The relative abundance of the top ten most abundant phyla, including Firmicutes, Bacteroidetes, Proteobacteria, and Actinobacteria, and the top ten most predominant genera, such as Faecalibacterium, Blautia, Bacteroides, and Bifidobacterium were identified in the two groups; the results for each group are shown in Figure 2A,C and in each sample in Appendix A. Higher levels of the phylum Firmicutes and the genera Faecalibacterium, Roseburia, and Bifidobacterium but a lower abundance of Bacteroidetes and Akkermansia were found in fecal samples from vaccinated subjects when compared to the unvaccinated group (Figure 2B,D). Overall, the two groups shared the vast majority of annotated bacteria, including 57 genera, such as Bifidobacterium, Akkermansia, Prevotella, Ruminococcus, Faecalibacterium, Butyricimonas, etc. (Figure 2E,F).Paraeggerthell is specific for vaccinated populations, but Catenibacteriu, Peptococcus, and Methanobrevibacter become extinct with the vaccine. We further assessed the F/B ratio, which has been reported to be associated with various diseases. A higher F/B ratio was detected in vaccinated subjects when compared to the unvaccinated group (Figure 2G; *P* = 0.03).

We compared the differentially abundant bacteria between groups and identified specific taxonomic bacteria that are more likely to distinguish vaccinated subjects from the controls. The association of relative bacterial abundances to vaccinated status was tested using LEfSe analysis. Microbial features with an LDA score >2 were regarded as markedly different. It was observed that Faecalibacterium and Mollicutes were significantly abundant in the vaccinated populations, whereas Prevotella, Enterococcus, Leuconostocaceae, and Weissella were negatively associated with the vaccination (Figure 3A,B). These results highlighted the fact that significant differences in gut microbiome existed between vaccinated individuals and controls, reinforcing the notion that vaccination elicited a distinct microbiome signature.

### 3.4. Enterotype Distribution Indicated an Inclination for Faecalibacterium-Dominated Types upon Vaccination

Next, we performed gut enterotype analysis to gain further insight into the characteristics of microbial communities in vaccinated and unvaccinated individuals. Forty samples were divided into three enterotypes using the PCA method according to the microbial compositions at the genus level (Figure 4A). The major contributors and most abundant genera in the three distinct gut enterotypes were Faecalibacterium, Blautia, and Prevotella, respectively. The relative abundances of these genera in each gut enterotype were confirmed in Figure 4B–D. It was found that 35% of the unvaccinated subjects were in the Faecalibacterium-predominant enterotype; 40% were in the Blautia enterotype; and 25% in the Prevotella enterotype (Figure 4E). In contrast, vaccinated subjects exhibited a higher proportion in Faecalibacterium (65%) and a lower percentage in Blautia (35%), but none in the Prevotella-dominant enterotype. The Blautia-enriched enterotype was composed of seven vaccinated and eight unvaccinated ones, whereas in the enterotype Prevotella, there were only the unvaccinated controls (Figure 4F).

### 3.5. Predicting Functional Capacities of Gut Microbiota Specific to Vaccinated Status via PICRUST Analysis

PICRUSt analysis was employed to examine the functions of gut microbiota following vaccination. A total of 271 KEGG pathways were generated based on the 16S rRNA sequencing data. The global functional features for each group were evaluated via β-diversity plots, as shown in Figure 5A–C and Appendix A. The plots of PCA, PCoA, and NMDS demonstrated dispersive clusters and clear separations (*P* = 0.003, Anosim) of intestinal bacterial functions in vaccinated and unvaccinated individuals, although no statistically significant difference was found in single axes (Appendix A). At pathway level 1, the fecal microbiota from vaccinated participants showed significantly depleted capacity related to human diseases when compared with the unvaccinated group (Figure 5D, *P* = 0.04). In the KEGG pathway level 2, a significant enrichment of pathways involving carbohydrate metabolism and secondary metabolite biosynthesis was observed in the vaccinated group as compared to unvaccinated controls (Figure 5E). On the other side, potential functions associated with neurodegenerative diseases, cardiovascular diseases, and cancers were depressed after vaccination. The shifts in microbial functions induced by vaccines might alleviate host susceptibility to SARS-CoV-2virus infection and confer considerable resistance against COVID-19.

## 4. Discussion

Previous studies have shown that changes in the gut microbiome are linked to COVID-19. It was recently reported that the intestinal microbiome in COVID-19 patients has a lower biodiversity when compared to healthy individuals. COVID-19 patients possess a decreased percentage of beneficial bacteria, such as Bifidobacteria adolescentis, but higher levels of opportunistic and pathogenic bacteria, such as Streptococcus anginosus [24]. In the current study, we included comprehensive profiling of the fecal microbiota from individuals receiving SARS-CoV-2 virus vaccines and unvaccinated subjects. Vaccinated subjects also showed distinct gut microbiome signatures with hallmark manifestations, including depressed diversity, elevated F/B ratios, distributional tendencies toward Faecalibacterium-predominant enterotypes, and altered gut microbiome compositions and functions, as compared with unvaccinated individuals. Specifically, fecal samples from SARS-CoV-2 virus vaccine recipients were identified to be enriched with Faecalibacterium and Mollicutes, along with a deficiency of Prevotella, Enterococcus, Leuconostocaceae, and Weissella. Finally, PICRUSt analysis further indicated the impact of SARS-CoV-2 vaccine inoculation on the elevation of the microbial potentials of KEGG pathways involving carbohydrate metabolism and transcription but depleted functions in neurodegenerative diseases, cardiovascular diseases, and cancers.

Currently, multiple lines of vaccines against COVID-19 have been tested. According to recent data, 3.44 billion doses of the COVID-19 vaccine have been administered to1.33 billion residents in China [25]. Because of the stable expression of conformation-dependent antigenic epitopes that are easily produced in large quantities [26], inactivated SARS-CoV-2 virus vaccines have been commonly used in China. BBIBP-CorV, an inactivated SARS-CoV-2 virus vaccine produced by the Beijing Institute of Biological Products Co., Ltd., has been approved for emergency vaccination of Chinese populations. Since the large-scale emergency use of BBIBP-CorV launched on 1 December 2020, the vaccination information of 49,7743 subjects has been collected and the safety of BBIBP-CorV evaluated, indicating that the overall incidence of adverse reactions was lower than 1.03% [27].

Gut microbiota has been proven to have a particularly essential role in maintaining host immune functions. Moreover, many studies have confirmed dysbiosis of gut microbiome structure and function in COVID-19 patients [28,29]. It has been suggested that the GI system is possibly involved in the pathogenesis of COVID-19, especially through the disturbances of the intestinal microbiome. The significant changes in the intestinal microbiome of patients with confirmed SARS-CoV-2 infection have been identified previously, with an enrichment of opportunistic pathogens and a depletion of beneficial commensals. It has been demonstrated that the abundance of Coprobacillus spp., Clostridium ramosum, and Clostridium hathewayiis correlated with the severity of COVID-19, but Faecalibacterium prausnitzii showed an inverse correlation. Bacteroides dorei, Bacteroides thetaiotaomicron, Bacteroides massiliensis, and Bacteroides ovatus, with potentials to reduce angiotensin-2-converting enzyme expression in the gut, were observed to be inversely correlated with the SARS-CoV-2 burden in feces [30]. However, the heterogeneity of the gut microbiome between unvaccinated and vaccinated individuals remains obscure. Thus, we conducted 16S rRNA sequencing of fecal samples in a cohort of 20 Chinese adults receiving two doses of BBIBP-CorV and 20 non-vaccinated individuals. We also examined the effect of the BBIBP-CorV vaccine on gut microbiota.

Bacterial diversity has been regarded as an important indicator of gut homeostasis [31]. This study found that vaccinated individuals exhibited microbial variations with significantly deficient OTUs and suppressed α-diversity indices. Interestingly, previous studies have highlighted the reduction in the diversity of bacterial communities in COVID-19 patients [28,32]. In some serious cases, microbial richness was not restored to normal levels even after a 6-month recovery [12]. Notably, we found that the degree of decline in the Chao1 index was lower in vaccinated individuals (median of Chao1 index in controls: 196; median of Chao1 index in vaccinated individuals: 158; degree of decline: 100%-158/196 = 19.39%) than in COVID-19 patients post-recovery (median of Chao1 index in healthy controls: 432, median of Chao1 index in post-convalescent patients: 259; degree of decline: 100%-259/432 = 40.05% in Chen Y et al. [12]). These findings provide evidence that gut bacterial diversity would significantly decrease, even when exposed to an inactivated SARS-CoV-2 virus vaccine such as BBIBP-CorV; however, the degree of decline appears to be relatively low. Other possibilities might be that a selected group of bacteria was promoted by vaccines, while others were either diminished or unchanged, thus establishing a healthier gut ecosystem.

Viral vaccines are designed to form an enduring memory of viral components for the adaptive immune system [33]. Vaccinated individuals are expected to be less susceptible to SARS-CoV-2 invasion and more resistant to COVID-19. In fact, our results indicated that the microbial composition in vaccinated recipients was indeed distinct from unvaccinated subjects. The human gut microbiota is mainly comprised of two major phyla, Bacteroidetes and Firmicutes [34]. In this study, a higher abundance of the phylum Firmicutes was found in vaccinated individuals, while more Bacteroidetes were detected in the unvaccinated group, thus leading to an increased F/B ratio in vaccinated subjects. Previous studies demonstrated a reduction in Firmicutes and an elevation in Bacteroidetes among patients with bipolar depression [35], major depressive disorders [36], or type 1 diabetes mellitus [37], supporting the necessity of a high F/B ratio for host health and better clinical outcomes; this is to some extent in agreement with our findings in vaccine recipients.

Moreover, a significant increase in the butyrate-producing bacteria Faecalibacterium [38] was identified in the vaccinated subjects. Butyrate, one of the major microbial fermentation products, serves as a crucial energy source of colonocytes [39] and has an essential role in immune accommodation, gut barrier regulation, gut metabolism, and energy modulation [38,39]. This study found that vaccinated individuals were enriched with the enterotype dominated by Faecalibacterium, which implies a possible enrichment of beneficial butyrate. To the contrary, a decreased abundance of several opportunistic pathogens, such as Enterococcus [32] and Prevotella [40], was not found in vaccinated patients. Recent studies suggested that elevated Prevotella in HIV is a driver for persistent inflammation in the gut, leading to mucosal dysfunction and systemic inflammation [41,42,43]. Furthermore, increased abundance of Prevotella has also been linked to obesity [44], hypertension [40], and non-alcoholic fatty liver disease [45]. The elimination of Prevotella might facilitate an improved gut environment following vaccination.

The intestinal epithelial barrier function and balanced gut microbiome are critical for gut immunity and metabolism. During COVID-19 infection, the virus was suspected to possibly disrupt the expression, distribution, and activity of intestinal transport proteins within cell membranes, such as the aldosterone-regulated epithelial sodium channel present in the distal colon [46]. More recently, a cross-sectional study revealed the altered gut microbiome caused by SARS-CoV-2 infection in patients with or without type 2 diabetes mellitus. More abundant Shigella, Bacteroides, and Megamonas were detected in COVID-19 patients with type 2 diabetes mellitus. Metabolic pathways including ribose transport system substrate-binding, bacterial/archaeal transporters, fructuronate reductase, GTP cyclohydrolase II, methenyltetrahydromethanopterin cyclohydrolase, lysozyme, and aspartate ammonia-lyase were enriched in the gut microbes of diabetes patients, while pathways such as copper resistance, D-galactarolactone cycloisomerase, alpha-galactosidase, DNA repair, crotonyl-CoA carboxylase/reductase, valine-pyruvate amino-transferase, cytidine2498-2′-O-methyltransferase, phosphoribosylformimino-5-aminoimidazole carboxamide, and a large subunit ribosomal protein were suppressed [47]. In this study, it was identified that, along with altered microbial compositions, multiple KEGG pathways in the aspect of microbial functions varied between groups. The results obtained from the PICRUSt analysis at KEGG levels indicated that bacterial potentials implicated in carbohydrate metabolism and transcription prospered with vaccination. Carbohydrate metabolism is known to have an important role in cellular energy and the biosynthesis of cellular building blockers [48]. Dysregulation of the carbohydrate metabolism is confirmed to be causative for various human diseases, such as diabetes mellitus and cancer [49]. On the other hand, vaccination conduces to depression of pathways involved in neurodegenerative diseases, cardiovascular diseases, and cancers. It has been evidenced that a reduction in these KEGG pathways, including cancers and neurodegenerative diseases, is implicated in high-intensity interval training-induced improvements in metabolic health [50].

Since the type of vaccine used in the present study was an inactivated vaccine but not a live attenuated vaccine, the potential mechanism by which the vaccine can alter the gut microbiome was pondered. Similarly, it was reported that an inactivated bivalent Aeromonas hydrophila/Aeromonas veronii vaccine significantly changed the structure, composition, and predictive function of intestinal mucosal microbiota, for example, by reducing the relative abundance of potential opportunistic pathogens [51]. Given the robust immune response directly stimulated by vaccination [52,53] and the fact that immune function plays a crucial role in maintaining mucosal microbial homeostasis, the inactivated vaccine might exert a dramatic effect on intestinal mucosal microbiota by enhancing immune function.

It had been previously known that the mechanism of SARS-CoV-2 cell entry is a crucial step during the initial stage of corona viruses infection. Aboshanab et al. have elucidated that SARS-CoV-2 might utilize the immunogenic studded spikes of glycoproteins on the surface of the virus as a pivotal factor to attach, fuse, and enter host cells such as enterocytes. It was suggested that by neutralizing antibodies targeting the receptor binding domain in viral S1 subunit proteins, small peptide inhibitors, peptide fusion inhibitors against S2 subunit proteins, host cell angiotensin converting enzymes 2, and protease inhibitors, the SARS-CoV-2 S protein interaction with host cell receptors might be significantly disrupted. This could be a potential course for controlling viral cell entrance [54]. Investigators have also suggested that gut microbiota dysbiosis is involved in the development and severity of COVID-19 symptoms by regulating SARS-CoV-2 entry [20]. Therefore, the intestinal flora might participate in the protection against SARS-CoV-2 entry in response to vaccination. In addition, epitope-based vaccines (EBVs) which harbor the least number of the optimally immunogenic epitopes, are believed to offer more effective and safe alternatives as compared to the conventional vaccines. Although candidate EBVs targeting SARS-CoV-2 comprising both B and T cell epitopes for concomitant induction of humoral and cellular immune responses have not yet been approved by the FDA, further examination of the impacts of EBVs on the gut microbiome [55] is worth while. It is recommended that future research focuses on the development of microbiota-based interventions for improving immune responses to SARS-CoV-2 vaccinations [20].

## 5. Conclusions

In conclusion, key microbial changes were proven to be vaccine-specific. We demonstrated significantly improved profiles and functional capacities of gut microbiota shaped by vaccination (BBIBP-CorV vaccine), which might have important implications for developing a robust immune barrier by the host. However, further studies are needed to elucidate the complex interactions between gut microbiota and SARS-CoV-2 virus vaccines.

## Figures and Tables

**Figure 1 vaccines-11-00942-f001:**
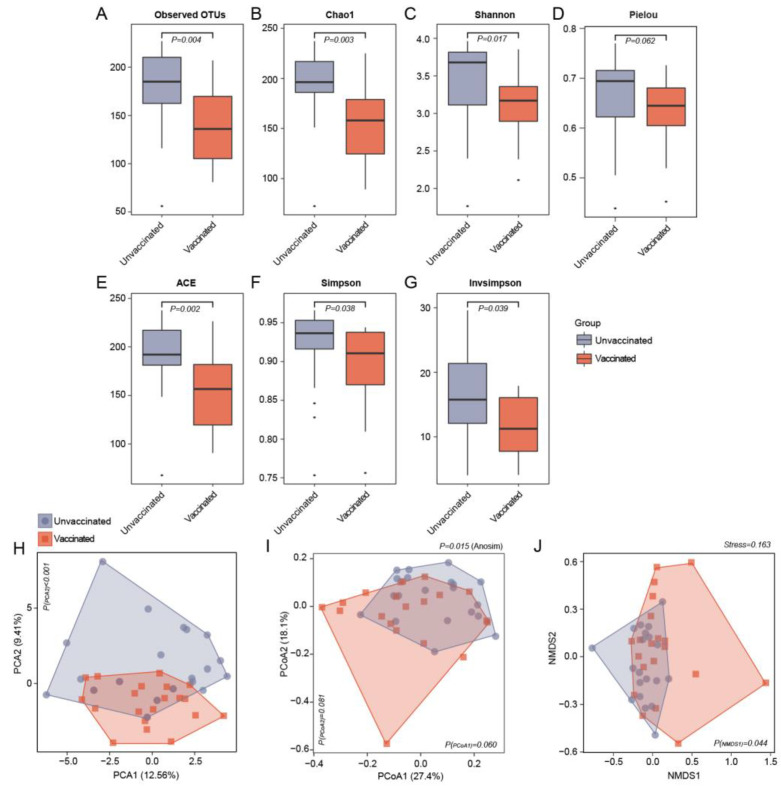
Gut microbial deviations of α-(within-sample) and β-(between-sample) diversity between vaccinated and unvaccinated participants. (**A**) Comparison of the number of observed OTUs in gut microbiota between vaccinated and unvaccinated participants. *P* = 0.004, Wilcoxon rank-sum test. (**B**–**G**) Indicators of microbial α-diversity at the genus level, including Chao1 index (**B**; *P* = 0.003), Shannon index (**C**; *P* = 0.017), Pielou evenness (**D**; *P* = 0.062), ACE indices (**E**; *P* = 0.002), Simpson indices (**F**; *P* = 0.038) and Invsimpson indices (**G**; *P* = 0.039) were assessed in unvaccinated and vaccinated individuals. The boxes represent the interquartile ranges; the line inside represents the median, and the points indicate outliers. *P* values are from Wilcoxon rank-sum test. (**H**–**J**) Scatter plots of β-diversity according to Bray–Curtis dissimilarities based on genus-level taxonomic profiles from 16S rRNA gene sequencing. PCA plots distinguishing vaccinated and unvaccinated subjects from each other (**H**; *P* value for PCA2 axis < 0.001, Wilcoxon rank-sum test). PCoA plots of samples in vaccinated and unvaccinated groups separate from each other (**I**; overall *P* = 0.015, Anosim test; *P* value for PCoA2 axis = 0.081, *P* value for PCoA1 axis = 0.06, Wilcoxon rank-sum test). Dispersion of each individual vaccinated or unvaccinated according to NMDS plots at the genus level (**J**; *P* value for NMDS1 axis = 0.044, Wilcoxon rank-sum test).

**Figure 2 vaccines-11-00942-f002:**
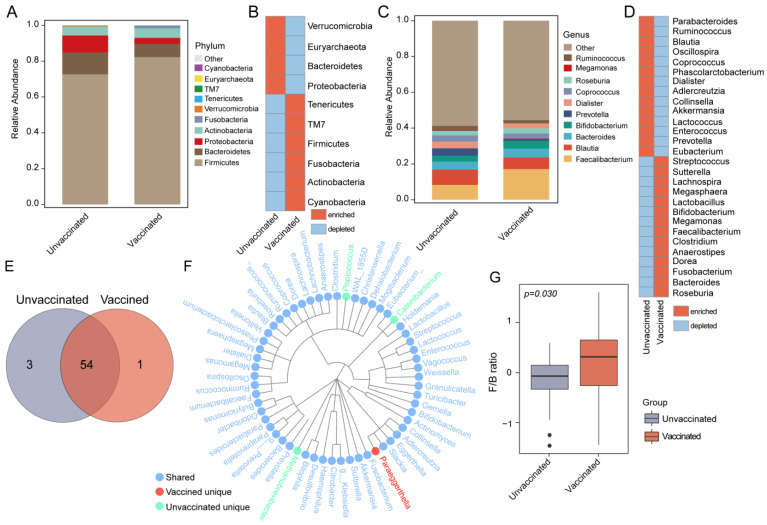
Taxonomic profiles of gut bacterial communities in vaccinated and unvaccinated individuals. (**A**) Phylum-level taxonomic abundance and proportion for each group. The top 10 most abundant phyla are annotated in the panel legend, and the remaining detected phyla are indicated as other. (**B**) Heat map illustrating the enrichment and depletion of the top 10 dominant phyla in the unvaccinated or vaccinated groups. (**C**) Bar plots indicating the relative abundance and proportion of the top 10 most abundant genera detected in the study cohort. Other indicates the sum of all the other genera except the top 10 genera. (**D**) Heat map depicting the top 30 most dominant genera enriched or depleted in the unvaccinated or vaccinated group. (**E**) Venn diagram showing the number of genera annotated in unvaccinated and vaccinated subjects. (**F**) Taxonomic tree for the genera shared or specific for unvaccinated and vaccinated individuals. Circles in blue indicate the shared genera between unvaccinated and vaccinated participants; circles in green indicate the unique genera in the unvaccinated group; circles in red indicate the unique genera in the vaccinated. (**G**) The ratio of F/B in the unvaccinated and vaccinated group (*P* = 0.034, Wilcoxon rank-sum test).

**Figure 3 vaccines-11-00942-f003:**
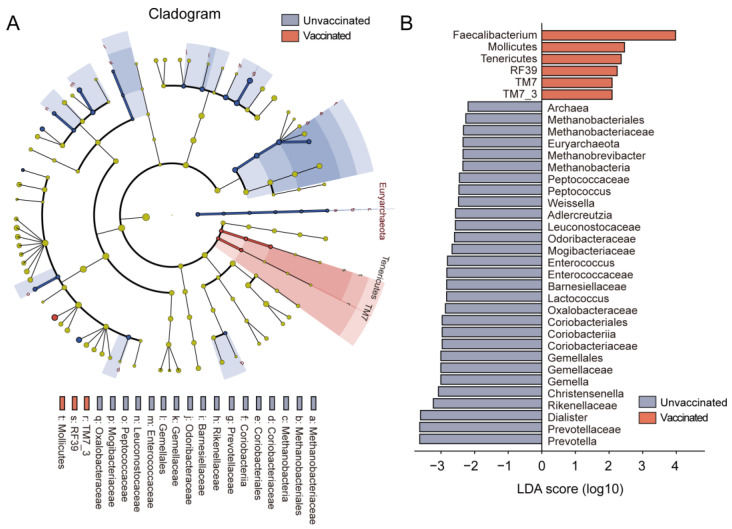
Fecal microbiome signatures and variations in vaccinated subjects as compared with unvaccinated by Lefse and LDA analyses. (**A**) Cladogram showing different taxonomic compositions in unvaccinated (blue) and vaccinated subjects (red). (**B**) Histogram of LDA scores showing differentially abundant taxa between unvaccinated (blue) and vaccinated subjects (red). The taxa with |LDA score (log10)| > 2 are listed.

**Figure 4 vaccines-11-00942-f004:**
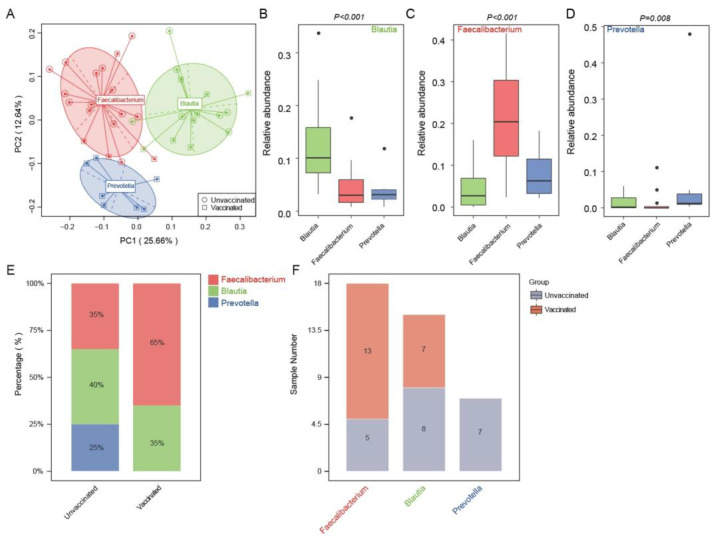
The discrepancy in gut enterotype features of vaccinated and unvaccinated participants. (**A**) A total of 40 samples from vaccinated and unvaccinated groups are stratified into three distinct gut enterotypes as visualized using PCA based on the genera. The major contributors in the three enterotypes are Faecalibacterium, Blautia, and Prevotella, respectively. (**B**–**D**) Relative abundance of the top genera (Blautia, Faecalibacterium, and Prevotella) in each enterotype. Boxes present the interquartile ranges; the inside lines represent the median, circles are outliers. *P* < 0.001 for Blautia abundance in enterotype Blautia, *P* < 0.001 for Faecalibacterium abundance in enterotype Faecalibacterium, *P* = 0.008 for Prevotella abundance in enterotype Prevotella, Kruskal–Wallis test. (**E**) The proportion of vaccinated and unvaccinated individuals distributed in the three enterotypes dominant in Blautia, Faecalibacterium, and Prevotella, respectively. (**F**) In the three distinct enterotypes, the number of subjects from unvaccinated and vaccinated groups is shown.

**Figure 5 vaccines-11-00942-f005:**
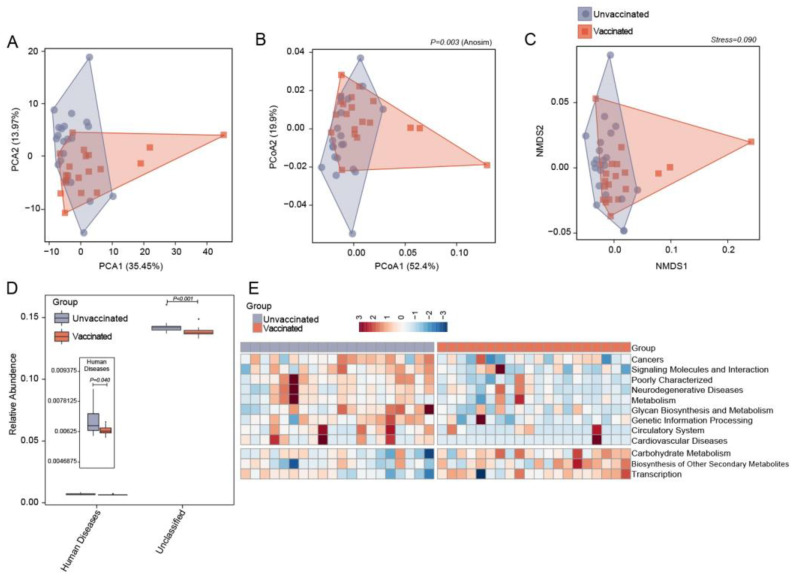
Gut microbial functional capabilities between unvaccinated and vaccinated individuals. (**A**–**C**) β-diversity assessment regarding microbial functions according to the relative abundance of KEGG pathways by PCA, PCoA, and NMDS scatter plots. *P* = 0.003, Anosim test. (**D**) Bar plots showing the KEGG pathways that are significantly different between unvaccinated (blue) and vaccinated (red) individuals in level 1. (**E**) Heat map for the relative abundance of significantly enriched KEGG pathways and depleted ones in vaccinated populations compared with unvaccinated ones on level 2. The abundance profiles are transformed into the Z scores by subtracting the average abundance and dividing the standard deviation of all samples; the score is negative in blue indicating when the abundance is lower than the mean, and positive in red indicating when it is higher.

**Table 1 vaccines-11-00942-t001:** Baseline characteristics of the unvaccinated and vaccinated participants.

	Unvaccinated (*n* = 20)	Vaccinated (*n* = 20)	*P* Value
Age, years	38.5 (29.5–49.0)	38.5 (30.5–42.5)	0.799
Male/Female	12/8	9/11	0.527
BMI, kg/m^2^	23.4 (20.2–25.5)	22.9 (21.3–26.5)	0.745
SBP, mmHg	120.5 (116.3–126.6)	120.0 (111.2–127.5)	0.489
DBP, mmHg	73.2 (70.0–80.0)	70.5 (65.3–77.3)	0.290
FBG, mmol/L	5.07 (4.63–5.39)	4.59 (4.38–5.30)	0.239
Total cholesterol, mmol/L	4.65 (3.83–5.12)	4.61 (4.11–5.24)	0.685
Triglyceride, mmol/L	1.29 (0.82–1.60)	0.98 (0.62–1.35)	0.074
HDLC, mmol/L	1.15 (0.92–1.57)	1.37 (1.02–1.70)	0.318
LDLC, mmol/L	2.55 (2.25–3.37)	2.83 (2.39–3.70)	0.196
Uric acid, umol/L	321.25 (233.98–386.50)	378.00 (243.75–407.25)	0.626
WBC, uL	6.95 (6.20–7.86)	5.89 (5.12–7.85)	0.062

Abbreviations: BMI, body mass index; SBP, systolic blood pressure; DBP, diastolic blood pressure; FBG, fasting blood glucose; HDLC, high-density lipoprotein cholesterol; LDLC, low-density lipoprotein cholesterol; WBC, white blood cell. *P* values are obtained from Wilcoxon test.

## Data Availability

The data set supporting the results of this article has been deposited in the EMBL European Nucleotide Archive (ENA) under BioProject accession codePRJEB48163 [http://www.ebi.ac.uk/ena/data/view/PRJEB48163 (accessed on 21 October 2021)].

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
