# Peer review of "BBIBP-CorV Vaccination against the SARS-CoV-2 Virus Affects the Gut Microbiome"

_vaccines, 2023, doi:10.3390/vaccines11050942_

Round 1
Reviewer 1 Report
First of all, I would like to congratulate the authors for submitting their quality work to this journal. The work reported by Shen et al in their research work entitled "Vaccination against the SARS-CoV-2 virus affects the gut microbiome". The work carried out by authors is very scientific and interesting and most needed for the time. The manuscript could be accepted in after minor revision. I have only a few suggestions:-
1. In Line no 24, in the abstract section, KEGG must be introduced for the first time.
2. In the introduction section, line no 66, we must be replaced with authors/investigators.
3. Why conclusion is written twice in section 5? Write one comprehensive conclusion.
Author Response
Dear reviewer,
Thank you very much for your consideration and encouragement on our article (Manuscript ID: vaccines-2359210). We thank the reviewers for their professional and positive comments to improve our manuscript. We have considered the comments, and revised the manuscript. A file that indicates the changes from our original submission as the file type marked up version of article-revised have been attached.
The point-by-point replies to reviewers are listed as below.
First of all, I would like to congratulate the authors for submitting their quality work to this journal. The work reported by Shen et al in their research work entitled "Vaccination against the SARS-CoV-2 virus affects the gut microbiome". The work carried out by authors is very scientific and interesting and most needed for the time. The manuscript could be accepted in after minor revision. I have only a few suggestions:-
- In Line no 24, in the abstract section, KEGG must be introduced for the first time.
Response: We thank the reviewer for this suggestion. "KEGG" in the abstract section has been described as "Kyoto Encyclopedia of Genes and Genomes" as suggested.
- In the introduction section, line no 66, we must be replaced with authors/investigators.
Response: Many thanks for this suggestion. "We" has been replaced with "investigators" in the introduction section mentioned above.
- Why conclusion is written twice in section 5? Write one comprehensive conclusion.
Response: Thanks for this suggestion. In section 5, the redundant conclusion section has been removed.
Reviewer 2 Report
Report
The present research vaccines-2359210 titled: “Vaccination against the SARS-CoV-2 virus affects the gut microbiome” was aimed at evaluating the effect of the BBIBP-CorV vaccine (ChiCTR2000032459, sponsored by the Beijing Institute of Biological Products/Sinopharm) on the altered gut microbiota. The topic is very interesting from the medical and environmental aspect, however; there are some important comments and suggestions that should be considered and fulfilled, and these are as follows:
1. The title of the manuscript is very broad since it is not reflecting the final study results and conclusion in terms of the type of vaccine used in this study, therefore I recommend including the type of vaccine used in this study to avoid generalization of COVID-9 vaccines. 2. Abbreviations should be first described at the first mention and then used consistently in the whole manuscript (examples, in the abstract, SARS-CoV-2, COVID-19, PICRUSt, KEGG. The whole manuscript should be thoroughly revised regarding this matter. 3. In the methodology, section, 2.1, the unvaccinated group, the authors should ensure that none of the individuals of this group has previously caught SARS Cov2 infection. Since, this is very important and should be verified using lab analysis such as serological measurement for example of SARS COV2- IgG and IgM, and has to be included in the methodology and results sections. It this very important to exclude these individuals from the study and prevent interference. 4. In the methodology, section, 2.1, the authors should include the number of the hospital committee ethics approval as well as provide the template of patient consent. 5. In the methodology, how did the author confirm that a participant has taken 2 doses of vaccination? Is done via oral communication with the participants, hospital records, or something else. 6. L99, the author should include the city, country(source) of Qubit@2 Fluorometer. 7. In the discussion section, it was previously known and published that Coronaviruses utilize the immunogenic studded spikes of glycoproteins on their surface as a key factor for attachment, fusion, and entrance to host cells such as enterocytes. This has to be included in the discussion section. (Exploring SARS-CoV-2 Spikes Glycoproteins for Designing Potential Antiviral Targets. Viral Immunol. 2021 Oct;34(8):510-521. doi: 10.1089/vim.2021.0023,. PMID: 34018828. 8. Also, in the discussion, The author should also discuss and elaborate on the potential mechanism by which the respective vaccine can alter the microbiota especially, since the type of vaccine used was inactivated vaccine and not a live-attenuated vaccine and administered vial IM and not orally. (Potential underlying mechanism 9. There are five recent, important, relevant, literatures that should be included and discussed in the discussion section to show up the differences and highlight the novel finding of this research: a. Maddah R, Goodarzi V, Asadi-Yousefabad SL, Abbasluo M, Shariati P, Shafiei Kafraj A. Evaluation of the gut microbiome associated with COVID-19. Inform Med Unlocked. 2023;38:101239. doi: 10.1016/j.imu.2023.101239. Epub 2023 Apr 3. PMID: 37033411; PMCID: PMC10069162. b. Lewandowski K, Kaniewska M, Rosołowski M, Rydzewska G. Gastrointestinal symptoms in COVID-19. Prz Gastroenterol. 2023;18(1):61-66. doi: 10.5114/pg.2021.112683. Epub 2022 Jan 18. PMID: 37007763; PMCID: PMC10050985. c. Sandle GI, Herod MR, Fontana J, Lippiat JD, Stockley PG. Is intestinal transport dysfunctional in COVID-19-related diarrhea? Am J Physiol Gastrointest Liver Physiol. 2023 Mar 28. doi: 10.1152/ajpgi.00021.2023. Epub ahead of print. PMID: 36976797. d. Hamed SM, Sakr MM, El-Housseiny GS, Wasfi R, Aboshanab KM. State of the art in epitope mapping and opportunities in COVID-19. Future Sci OA. 2023 Feb;16(3-06):FSO832. doi: 10.2144/fsoa-2022-0048. Epub 2023 Mar 6. PMID: 36897962; PMCID: PMC9987558. e. Mannan A, Hoque MN, Noyon SH, Hamidullah Mehedi HM, Foisal J, Salauddin A, Rafiqul Islam SM, Sharmen F, Tanni AA, Siddiki AZ, Tay A, Siddique M, Shaminur Rahman M, Galib SM, Akter F. SARS-CoV-2 infection alters the gut microbiome in diabetes patients: A cross-sectional study from Bangladesh. J Med Virol. 2023 Mar 22. doi: 10.1002/jmv.28691. Epub ahead of print. PMID: 36946508. 10. Conclusion section is repeated two times???!!. 11. The reference section needs to be updated with relevant and updated literature since many similar studies have been conducted recently and are not reflected in this research (see above relevant reference). The most recent reference is published in 2020 therefore relevant and updated literature published in 2021-2023 should be included in either the introduction or in the discussion section.
Therefore, and for the above-mentioned remarks, I advised a revision of the respective manuscript in its current state taking into consideration the above comments and recommendations before being considered for publication

Report
The present research vaccines-2359210 titled: “Vaccination against the SARS-CoV-2 virus affects the gut microbiome” was aimed at evaluating the effect of the BBIBP-CorV vaccine (ChiCTR2000032459, sponsored by the Beijing Institute of Biological Products/Sinopharm) on the altered gut microbiota. The topic is very interesting from the medical and environmental aspect, however; there are some important comments and suggestions that should be considered and fulfilled, and these are as follows:
1. The title of the manuscript is very broad since it is not reflecting the final study results and conclusion in terms of the type of vaccine used in this study, therefore I recommend including the type of vaccine used in this study to avoid generalization of COVID-9 vaccines. 2. Abbreviations should be first described at the first mention and then used consistently in the whole manuscript (examples, in the abstract, SARS-CoV-2, COVID-19, PICRUSt, KEGG. The whole manuscript should be thoroughly revised regarding this matter. 3. In the methodology, section, 2.1, the unvaccinated group, the authors should ensure that none of the individuals of this group has previously caught SARS Cov2 infection. Since, this is very important and should be verified using lab analysis such as serological measurement for example of SARS COV2- IgG and IgM, and has to be included in the methodology and results sections. It this very important to exclude these individuals from the study and prevent interference. 4. In the methodology, section, 2.1, the authors should include the number of the hospital committee ethics approval as well as provide the template of patient consent. 5. In the methodology, how did the author confirm that a participant has taken 2 doses of vaccination? Is done via oral communication with the participants, hospital records, or something else. 6. L99, the author should include the city, country(source) of Qubit@2 Fluorometer. 7. In the discussion section, it was previously known and published that Coronaviruses utilize the immunogenic studded spikes of glycoproteins on their surface as a key factor for attachment, fusion, and entrance to host cells such as enterocytes. This has to be included in the discussion section. (Exploring SARS-CoV-2 Spikes Glycoproteins for Designing Potential Antiviral Targets. Viral Immunol. 2021 Oct;34(8):510-521. doi: 10.1089/vim.2021.0023,. PMID: 34018828. 8. Also, in the discussion, The author should also discuss and elaborate on the potential mechanism by which the respective vaccine can alter the microbiota especially, since the type of vaccine used was inactivated vaccine and not a live-attenuated vaccine and administered vial IM and not orally. (Potential underlying mechanism 9. There are five recent, important, relevant, literatures that should be included and discussed in the discussion section to show up the differences and highlight the novel finding of this research: a. Maddah R, Goodarzi V, Asadi-Yousefabad SL, Abbasluo M, Shariati P, Shafiei Kafraj A. Evaluation of the gut microbiome associated with COVID-19. Inform Med Unlocked. 2023;38:101239. doi: 10.1016/j.imu.2023.101239. Epub 2023 Apr 3. PMID: 37033411; PMCID: PMC10069162. b. Lewandowski K, Kaniewska M, Rosołowski M, Rydzewska G. Gastrointestinal symptoms in COVID-19. Prz Gastroenterol. 2023;18(1):61-66. doi: 10.5114/pg.2021.112683. Epub 2022 Jan 18. PMID: 37007763; PMCID: PMC10050985. c. Sandle GI, Herod MR, Fontana J, Lippiat JD, Stockley PG. Is intestinal transport dysfunctional in COVID-19-related diarrhea? Am J Physiol Gastrointest Liver Physiol. 2023 Mar 28. doi: 10.1152/ajpgi.00021.2023. Epub ahead of print. PMID: 36976797. d. Hamed SM, Sakr MM, El-Housseiny GS, Wasfi R, Aboshanab KM. State of the art in epitope mapping and opportunities in COVID-19. Future Sci OA. 2023 Feb;16(3-06):FSO832. doi: 10.2144/fsoa-2022-0048. Epub 2023 Mar 6. PMID: 36897962; PMCID: PMC9987558. e. Mannan A, Hoque MN, Noyon SH, Hamidullah Mehedi HM, Foisal J, Salauddin A, Rafiqul Islam SM, Sharmen F, Tanni AA, Siddiki AZ, Tay A, Siddique M, Shaminur Rahman M, Galib SM, Akter F. SARS-CoV-2 infection alters the gut microbiome in diabetes patients: A cross-sectional study from Bangladesh. J Med Virol. 2023 Mar 22. doi: 10.1002/jmv.28691. Epub ahead of print. PMID: 36946508. 10. Conclusion section is repeated two times???!!. 11. The reference section needs to be updated with relevant and updated literature since many similar studies have been conducted recently and are not reflected in this research (see above relevant reference). The most recent reference is published in 2020 therefore relevant and updated literature published in 2021-2023 should be included in either the introduction or in the discussion section.
Therefore, and for the above-mentioned remarks, I advised a revision of the respective manuscript in its current state taking into consideration the above comments and recommendations before being considered for publication
Author Response
Dear reviewer,
Thank you very much for your consideration and encouragement on our article (Manuscript ID: vaccines-2359210). We thank the reviewers for their professional and positive comments to improve our manuscript. We have considered the comments, and revised the manuscript. A file that indicates the changes from our original submission as the file type marked up version of article-revised have been attached.
The point-by-point replies to reviewers are listed as below.
The present research vaccines-2359210 titled: “Vaccination against the SARS-CoV-2 virus affects the gut microbiome” was aimed at evaluating the effect of the BBIBP-CorV vaccine (ChiCTR2000032459, sponsored by the Beijing Institute of Biological Products/Sinopharm) on the altered gut microbiota. The topic is very interesting from the medical and environmental aspect, however; there are some important comments and suggestions that should be considered and fulfilled, and these are as follows:
- The title of the manuscript is very broad since it is not reflecting the final study results and conclusion in terms of the type of vaccine used in this study, therefore I recommend including the type of vaccine used in this study to avoid generalization of COVID-9 vaccines.
Response: We thank the reviewer for this suggestion. The type of vaccine used in this study has been included in the title of the manuscript as suggested.
- Abbreviations should be first described at the first mention and then used consistently in the whole manuscript (examples, in the abstract, SARS-CoV-2, COVID-19, PICRUSt, KEGG. The whole manuscript should be thoroughly revised regarding this matter.
Response: Many thanks for these suggestions. The abbreviations in the whole manuscript have been described at the first mention as suggested.
- In the methodology, section, 2.1, the unvaccinated group, the authors should ensure that none of the individuals of this group has previously caught SARS Cov2 infection. Since, this is very important and should be verified using lab analysis such as serological measurement for example of SARS COV2- IgG and IgM, and has to be included in the methodology and results sections. It this very important to exclude these individuals from the study and prevent interference.
Response: Thanks for this suggestion. Individuals who have previously caught SARS Cov2 infection were strictly excluded in the current study. As serum samples were not simultaneously harvested when the study was carried out, it is difficult for us to perform additional serological measurement of SARS COV2- IgG and IgM. Nevertheless, we have confirmed that all the participants were free from SARS Cov2 infection by examining all their previous nucleic acid testing results for COVID-19 of throat swabs. Each of the participants subjected to nucleic acid testing for COVID-19 at a frequency of <72 hours per test. These have been described more clearly in the methodology, section, 2.1.
- In the methodology, section, 2.1, the authors should include the number of the hospital committee ethics approval as well as provide the template of patient consent.
Response: We thank the reviewer for this suggestion. The number of the hospital committee ethics approval has been added in the methodology, section, 2.1. And the template of patient consent has been attached.
- In the methodology, how did the author confirm that a participant has taken 2 doses of vaccination? Is done via oral communication with the participants, hospital records, or something else.
Response: Thanks for this advice. The vaccination institutions have provided a certificate recording the type, dose, inject date and manufacturer of vaccines for each individual. The vaccination history of the participants was confirmed according to the vaccination certificate. These have been described more clearly in the methodology, section, 2.1.
- L99, the author should include the city, country (source) of Qubit@2 Fluorometer.
Response: We thank the reviewer for this suggestion. The city, country (source) of Qubit@2 Fluorometer have been added in the methodology, section, 2.3.
- In the discussion section, it was previously known and published that Coronaviruses utilize the immunogenic studded spikes of glycoproteins on their surface as a key factor for attachment, fusion, and entrance to host cells such as enterocytes. This has to be included in the discussion section. (Exploring SARS-CoV-2 Spikes Glycoproteins for Designing Potential Antiviral Targets. Viral Immunol. 2021 Oct;34(8):510-521. doi: 10.1089/vim.2021.0023,. PMID: 34018828.
Response: Many thanks for this kind suggestion. The published reference regarding exploration of SARS-CoV-2 spikes glycoproteins for designing potential antiviral targets has been discussed in the revised manuscript at the discussion section as suggested. It is previously known that the mechanism of SARS-CoV-2 cell entry is a quite crucial step during the initial stage of coronaviruses infection. Aboshanab et al. have elucidated that SARS-CoV-2 might utilize the immunogenic studded spikes of glycoproteins on virus surface as a pivotal factor to attach, fuse, and enter into host cells such as enterocytes. It was thus suggested that by neutralizing antibodies targeting receptor binding domain in viral S1 subunit proteins, small peptide inhibitors, peptide fusion inhibitors against S2 subunit proteins, host cell angiotensin converting enzymes 2, and protease inhibitors, the SARS-CoV-2 S protein interaction with host cell receptors might be significantly disrupted, which might be a potential way for controlling viral cell entrance (doi: 10.1089/vim.2021.0023). Investigators have also suggested that gut microbiota dysbiosis is involved in the development and severity of COVID-19 symptoms by regulating SARS-CoV-2 entry (10.3390/microorganisms11020452). Therefore, the intestinal flora in response to vaccination might participate in the protection against SARS-CoV-2 entry. These have been discussed in the manuscript at Page 13 Lines 455 (marked up version).
- Also, in the discussion, The author should also discuss and elaborate on the potential mechanism by which the respective vaccine can alter the microbiota especially, since the type of vaccine used was inactivated vaccine and not a live-attenuated vaccine and administered vial IM and not orally. (Potential underlying mechanism).
Response: Thanks for this suggestion. Since the type of vaccine used in the present work was inactivated vaccine but not a live-attenuated vaccine, the potential mechanism by which the vaccine can alter the gut microbiome was wondered. Similarly, it was reported that, an inactivated bivalent Aeromonas hydrophila/Aeromonas veronii vaccine significantly changed the structure, composition, and predictive function of intestinal mucosal microbiota, for example, by reducing the relative abundance of potential opportunistic pathogens (10.1186/s40168-022-01409-6). Given the robust immune response directly stimulated by vaccination (10.1371/journal.pone.0263468, 10.1007/s11357-021-00471-6), and immune function plays a crucial role in maintaining mucosal microbial homeostasis, the inactivated vaccine might exert dramatic effect on intestinal mucosal microbiota by enhancing immune function. These have been discussed in the manuscript at Page 13 Lines 445 (marked up version).
- There are five recent, important, relevant, literatures that should be included and discussed in the discussion section to show up the differences and highlight the novel finding of this research: Maddah R, Goodarzi V, Asadi-Yousefabad SL, Abbasluo M, Shariati P, Shafiei Kafraj A. Evaluation of the gut microbiome associated with COVID-19. Inform Med Unlocked. 2023;38:101239. doi: 10.1016/j.imu.2023.101239. Epub 2023 Apr 3. PMID: 37033411; PMCID: PMC10069162. b. Lewandowski K, Kaniewska M, Rosołowski M, Rydzewska G. Gastrointestinal symptoms in COVID-19. Prz Gastroenterol. 2023;18(1):61-66. doi: 10.5114/pg.2021.112683. Epub 2022 Jan 18. PMID: 37007763; PMCID: PMC10050985. c. Sandle GI, Herod MR, Fontana J, Lippiat JD, Stockley PG. Is intestinal transport dysfunctional in COVID-19-related diarrhea? Am J Physiol Gastrointest Liver Physiol. 2023 Mar 28. doi: 10.1152/ajpgi.00021.2023. Epub ahead of print. PMID: 36976797. d. Hamed SM, Sakr MM, El-Housseiny GS, Wasfi R, Aboshanab KM. State of the art in epitope mapping and opportunities in COVID-19. Future Sci OA. 2023 Feb;16(3-06):FSO832. doi: 10.2144/fsoa-2022-0048. Epub 2023 Mar 6. PMID: 36897962; PMCID: PMC9987558. e. Mannan A, Hoque MN, Noyon SH, Hamidullah Mehedi HM, Foisal J, Salauddin A, Rafiqul Islam SM, Sharmen F, Tanni AA, Siddiki AZ, Tay A, Siddique M, Shaminur Rahman M, Galib SM, Akter F. SARS-CoV-2 infection alters the gut microbiome in diabetes patients: A cross-sectional study from Bangladesh. J Med Virol. 2023 Mar 22. doi: 10.1002/jmv.28691. Epub ahead of print. PMID: 36946508.
Response: We thank the reviewer for this suggestion. The above recent, important and relevant references have been added and discussed in the discussion section.
1) It was recently reported that the intestinal microbiome in COVID-19 patients has a lower biodiversity compared to healthy individuals. COVID-19 patients possess a decreased percentage of beneficial bacteria such as bifidobacteria adolescentis, and in contrast higher levels of opportunistic and pathogenic bacteria such as Streptococcus anginosus (doi: 10.1016/j.imu.2023.101239). These have been discussed in the manuscript at Page 11 Lines 331 (marked up version).
2) The gastrointestinal (GI) system has been suggested to be possibly involved in the pathogenesis of COVID-19, especially through the disturbances of the intestinal microbiome. The significant changes in the intestinal microbiome of 15 patients with confirmed SARS-CoV-2 infection have been identified previously, with an enrichment of opportunistic pathogens and a depletion of beneficial commensals. The abundance of Coprobacillus spp., Clostridium ramosum, and Clostridium hathewayi were demonstrated to be correlated with the severity of COVID-19, but Faecalibacterium prausnitzii showed an inverse correlation. Bacteroides dorei, Bacteroides thetaiotaomicron, Bacteroides massiliensis, and Bacteroides ovatus, with potentials to reduce angiotensin-2-converting enzyme expression in the gut were observed to be inversely correlated with SARS-CoV-2 burden in the faeces (doi: 10.5114/pg.2021.112683). These have been discussed in the manuscript at Page 12 Lines 360 (marked up version).
3) The intestinal epithelial barrier function and balanced gut microbiome are critical for gut immunity and metabolism. During COVID-19 infection, the virus was suspected to possibly disrupts the expression, distribution and activity of intestinal transport proteins within cell membranes, such as aldosterone-regulated epithelial sodium channel present in distal colon (doi: 10.1152/ajpgi.00021.2023). These have been discussed in the manuscript at Page 13 Lines 418 (marked up version).
4) Rpitope-based vaccines (EBVs), which harbor the least number of the optimally immunogenic epitopes, are believed to offer more effective and safe alternatives as compared to the conventional vaccines. Although candidate EBVs targeting SARS-CoV-2 comprising both B- and T-cell epitopes for concomitant induction of humoral and cellular immune responses have not yet been approved by the FDA, it worth further examination for EBVs' impacts on the gut microbiome (doi: 10.2144/fsoa-2022-0048). These have been discussed in the manuscript at Page 14 Lines 467 (marked up version).
5) Most recently, a cross-sectional study revealed the altered gut microbiome caused by SARS-CoV-2 infection in patients with or without type 2 diabetes mellitus. More abundant Shigella, Bacteroides, and Megamonas were detected in COVID-19 patients with type 2 diabetes mellitus. Metabolic pathways including ribose transport system substrate-binding, bacterial/archaeal transporters, fructuronate reductase, GTP cyclohydrolase II, methenyltetrahydromethanopterin cyclohydrolase, lysozyme, and aspartate ammonia-lyase were enriched in the microbes of diabetes patient's gut, while pathways such as copper resistance, D-galactarolactone cycloisomerase, alpha-galactosidase, DNA repair, crotonyl-CoA carboxylase/reductase, valine-pyruvate amino-transferase, cytidine2498-2′-O-methyltransferase, phosphoribosylformimino-5-aminoimidazole carboxamide, large subunit ribosomal protein were suppressed (doi: 10.1002/jmv.28691). These have been discussed in the manuscript at Page 13 Lines 422 (marked up version).
- Conclusion section is repeated two times???!!.
Response: Thanks for this advice. In section 5, the redundant conclusion section has been removed.
- The reference section needs to be updated with relevant and updated literature since many similar studies have been conducted recently and are not reflected in this research (see above relevant reference). The most recent reference is published in 2020 therefore relevant and updated literature published in 2021-2023 should be included in either the introduction or in the discussion section.
Response: Many thanks for these suggestions. The above recent, important and relevant references mentioned in Q9 have been added and discussed in detail in the discussion section. In addition, in the introduction, a few relevant and updated literatures published during 2021-2023 have been included at Page 2 Lines 47, 66, 72 (marked up version). Researchers have made efforts to decode the bidirectional relationship between gut microbiome and COVID-19, and disturbance in the gut composition is suggested to increase the susceptibility to COVID-19 (10.1016/j.heliyon.2023.e13801). The safety and efficacy of Sinopharm vaccine (BBIBP-CorV) have been verified in both elderly population and children (10.1136/postmj/postgradmedj-2022-141649, 10.1136/bmj-2022-073070). A correlation of gut microbiota and metabolic functions with the antibody response to the BBIBP-CorV vaccine has been documented recently. It was found that several short-chain fatty acids displayed a positive correlation with the antibody response (10.1016/j.xcrm.2022.100752). Meanwhile, preexisting antibodies targeting SARS-CoV-2 S2 was demonstrated to cross-react with gut bacteria and further impact the immunity induced by COVID-19 vaccine (10.1080/19490976.2022.2117503). It was inferred that gut microbiota possibly play a role in influencing the immune responses to COVID-19 vaccinations via mechanisms including effects of lipopolysaccharides, flagellin, peptidoglycan, and short-chain fatty acids (10.3390/microorganisms11020452). Future research directions are recommended to focus on the development of microbiota-based interventions on improving immune response to SARS-CoV-2 vaccinations (10.3390/microorganisms11020452).
Reviewer 3 Report
The reviewer immensely appreciates the authors approach to assess the DNA extracted from faecal samples through 16S ribosomal RNA sequencing analysis. Through there are few studies on gut microbiome in association to COVID-19 vaccination, but this study has its own merit and informative.
The author can carry out few suggestions and spelling checks. I recommend author for minor corrections.
# Line 142 : Spelling Check ..it should be subjects
#The subjects' history is important so try to display it on main section of the manuscript rather than in the supplementary files.
# Figure 2 F, try to get high resolution image.
Author Response
Dear reviewer,
Thank you very much for your consideration and encouragement on our article (Manuscript ID: vaccines-2359210). We thank the reviewers for their professional and positive comments to improve our manuscript. We have considered the comments, and revised the manuscript. A file that indicates the changes from our original submission as the file type marked up version of article-revised have been attached.
The point-by-point replies to reviewers are listed as below.
The reviewer immensely appreciates the authors approach to assess the DNA extracted from faecal samples through 16S ribosomal RNA sequencing analysis. Through there are few studies on gut microbiome in association to COVID-19 vaccination, but this study has its own merit and informative.
The author can carry out few suggestions and spelling checks. I recommend author for minor corrections.
- Line 142 : Spelling Check ..it should be subjects.
Response: We thank the reviewer for this suggestion. The spelling has been corrected.
- The subjects' history is important so try to display it on main section of the manuscript rather than in the supplementary files.
Response: Many thanks for this suggestion. The baseline characteristics of the participants unvaccinated or vaccinated in Table S1 has been moved into the main manuscript as Table 1.
- Figure 2 F, try to get high resolution image.
Response: Thanks for this suggestion. Figure 2 F has been replaced with high resolution image.